**Data Availability Statement:** An ethics committee has placed an ethical restriction on sharing de-identified data, since the data may contain sensitive

# Association between depressive symptoms and objective/subjective socioeconomic status among older adults of two regions in Myanmar

**Yuri Sasaki**[1]*, **Yugo Shobugawa**[2], **Ikuma Nozaki**[3], **Daisuke Takagi**[4], **Yuiko Nagamine**[5], **Masafumi Funato**[6], **Yuki Chihara**[2], **Yuki Shirakura**[2], **Kay Thi Lwin**[7], **Poe Ei Zin**[7], **Thae Zarchi Bo**[7], **Tomofumi Sone**[1], **Hla Hla Win**[7,8]

1 National Institute of Public Health, Wako, Japan, 2 Niigata University Graduate School of Medical and Dental Sciences, Niigata, Japan, 3 National Center for Global Health and Medicine, Tokyo, Japan, 4 The University of Tokyo, Tokyo, Japan, 5 Tokyo Medical and Dental University, Tokyo, Japan, 6 Harvard T.H. Chan School of Public Health, Boston, Massachusetts, United States of America, 7 University of Medicine 1, Yangon, Myanmar, 8 University of Public Health, Yangon, Myanmar

* sasakiy1006@gmail.com

## Abstract

Low objective socioeconomic status (SES) has been correlated with poor physical and mental health among older adults. Some studies suggest that subjective SES is also important for ensuring sound physical and mental health among older adults. However, few studies have been conducted on the impact of both objective and subjective SES on mental health among older adults. This study examines whether objective or subjective SES is associated with depressive symptoms in older adults in Myanmar. This cross-sectional study, conducted between September and December, 2018, used a multistage sampling method to recruit participants from two regions of Myanmar, for face-to-face interviews. The Geriatric Depression Scale (GDS) was used to evaluate the depressive symptoms. Participants were classified as having no depressive symptom (GDS score <5) and having depressive symptoms (GDS score ≥5). Objective and subjective SES were assessed using the wealth index and asking participants a multiple-choice question about their current financial situation, respectively. The relationship between objective/subjective SES and depressive symptoms was examined using a multivariable logistic regression analysis. The mean age of the 1,186 participants aged 60 years and above was 69.7 (SD: 7.3), and 706 (59.5%) were female. Among them, 265 (22.3%) had depressive symptoms. After adjusting for objective SES and other covariates, only low subjective SES was positively associated with depressive symptoms (adjusted odds ratio, AOR: 4.18, 95% confidence interval, CI: 2.98–5.87). This association was stronger among participants in the rural areas (urban areas, AOR: 2.10, 95% CI: 1.08–4.05; rural areas, AOR: 5.65, 95% CI: 3.69–8.64). Subjective SES has a stronger association with depressive symptoms than objective SES, among older adults of the two regions in Myanmar, especially in the rural areas. Interventions for depression in older adults should consider regional differences in the context of subjective SES by reducing socioeconomic disparities among the communities.

information regarding the respondents' physical and mental health. The following is the contact information of the individuals responsible for the dataset. Ethical Review Committee of the National Institute of Public Health, Japan TEL +81-48-458-6111 FAX +81-48-469-1573 2-3-6 Minami, Wako City, Saitama, Japan, 351-0197

**Funding:** The research was funded by AMED (Japan Agency for Medical Research and Development), for the project titled, "Development of a Health-Equity Assessment Tool Based on Social Epidemiological Survey for Older Adults in Myanmar and Malaysia" (17934739), Grants in aid for Scientific Research from the Japan Society for the Promotion of Science, for the project titled, "Differences in Social Capital Influence on Depression among Older People—A Comparative Study of Three Asian Countries" (19K19472), and Grants in aid for Health and Labor Administration Promotion Research Project titled 'Study on promotion of active and healthy aging in ASEAN (20BA2002)'. In addition, the World Health Organization for Health Development funded the research as research to accelerate universal health coverage in light of population ageing in ASEAN countries for research titled, "Development and Validation of Questionnaire Instrument for Evaluating the Determinants of Health Status and Universal Health Coverage in Older Adults in Selected Population in Myanmar and Malaysia (WHO Kobe Centre - WKC: K18015)". The funders had no role in study design, data collection and analysis, decision to publish, or preparation of the manuscript.

**Competing interests:** The authors have declared that no competing interests exist.

## Introduction

Objective socioeconomic status (SES) is defined as the economic and social position such as working status, household wealth, and poverty status [1–4]. Previous studies have recognized that low objective SES correlates with poor physical and mental health in older adults [4–8]. It also relates to daily care needs, and long-term care needs [4, 5, 7, 9]. Studies also indicate that older adults with a low income were more likely to be at a higher risk of diabetes [10], be functionally dependent, and use more inpatient and health care services [8]. Objective SES also influences mental health, and significant and positive relationships have been observed between objective SES and mental health of older adults in China, the United States, and Japan [5, 6, 9].

Subjective SES is defined as a person's conception of his or her position compared with that of others. For example, if a respondent rated him/herself as lower than middle class in the country or the community, the respondent is defined as having a lower SES [1–3, 11–13]. In a literature review on subjective SES and health, lower subjective SES was associated with significantly increased odds of non-communicable diseases, with a trend toward increased odds of obesity [13]. It suggested that the perception of one's own status in a social hierarchy has effects on one's health. This is in line with studies of Japanese older adults, whose subjective ratings of SES predict poor subjective mental health at a similar level of objective SES indicators [11]. Low subjective childhood SES also has a long-latency effect on the onset of depression among Japanese older adults [14]. Additionally, subjective social status (SSS), a concept similar to subjective SES [15] has been found to mediate the associations between objective SES and depression [16]. Moreover, some studies have found associations between objective/subjective SES and health measures such as self-rated health and life satisfaction, among older adults in Korea [17] and Taiwan [18]. One meta-analysis revealed a significant independent association between the subjective SES and physical health in adults, beyond traditional objective indicators of SES such as education, occupation, and income [19]. Therefore, both objective and subjective SES are hypothesized to indicate the most significant disparities, such that individuals with lower SES tend to have poor physical and psychological health. However, this relationship does not seem to simply reflect the effects of poverty [19]. Evidence of the association between SES (measured in various forms) and health has been interpreted as evidence that social stratification, not simply objective socioeconomic resources, have a meaningful impact on physical health [19, 20].

However, there are few studies regarding the effects of objective/subjective SES on depressive symptoms among older adults in developing countries including Myanmar. However, depressive symptoms are becoming more common among older adults with the increase of the aging population, especially in Asian countries [21]. The socioeconomic cost for older adults is vast due to higher rates of morbidity and mortality, and increased health care utilization and economic cost, compared to younger adults [22]. In Myanmar, the proportion of the population aged $\geq 60$ years is anticipated to increase from approximately 10% in 2020, to proximately 18.5% by 2050 [23]. The proportion of adults with depression is also expected to increase in the near future [24]. Depression and anxiety account for 5% of disability-adjusted life years, which puts them in the top 10 contributors of disability in Myanmar [24, 25]. However, there is no medical policy issued by public medical organization regarding mental health in Myanmar. As a result, mental health services have not received priority in primary health care, preventing thousands of people from accessing the mental health services they need [24]. Studies in Myanmar have found that depressive symptoms in older adults were strongly associated with diminished independence in performing seven functions (walking, ascending and descending stairs, feeding, dressing, going to the toilet, bathing, and grooming), a lower quality

of life [26], and lower economic and health status [27]. Several health organizations provide community-based health care services, even in remote areas, and are seeking to coordinate health service provision with the central health care system [28]. Religious organizations are also involved in service provision, and their role is gaining importance with the increasing need for collaborative action in the domain of health [29]. However, effective medical care systems including mental health services are still underdeveloped. Further, few studies regarding the mental health of older adults have been published in Myanmar. These are due to the international isolation of the country under military control for several years, during which the national health investment was found to be very low [30–32]. This study aimed to investigate whether objective/subjective SES, which are examined in the same model, are associated with depressive symptoms in older adults in two regions of Myanmar.

## Materials and methods

### Study design and participants

This study used a cross-sectional baseline survey for a longitudinal study. It was conducted between September and December, 2018 in two regions of Myanmar, and examined the predictors of physical and psychological health of 1,200 community-dwelling older adults aged ≥ 60 years. The target populations were those in the urban area of the Yangon region and the rural area of the Bago region, 91 kilometers northeast of Yangon.

A multistage random sampling method was used to select participants from the two regions. There are 45 townships in the Yangon region and 28 in the Bago region. First, six townships were randomly selected from each region, based on population proportionate sampling [33]. Following this, in Yangon, 10 wards were further randomly selected from each township, while in Bago, 10 village tracts were selected from each township, based on the population of each township/village tract. Finally, 10 people were randomly selected from each ward/village tract using the ledger lists of residents aged 60 years or older. In rural areas, there are multiple villages within a single village tract. In such cases, one of the villages was randomly selected from the village tract.

The difference between a ward and village tract is the degree of urbanization. The ward is the minimum unit of a residential district in an urban area, and the village tract is the corresponding level in rural areas. However, wards and village tracts sometimes co-exist within a township. In this survey, we selected only wards from Yangon and only village tracts from Bago to capture the features of urban and rural areas from each region. We considered Yangon as representative of urban areas and Bago as that of rural areas.

Trained surveyors visited homes of the residents with a public health nurse to meet participants. The inclusion criteria were individuals aged 60 years or older who were residents of the selected ward or village tract. We excluded individuals who were bed-ridden or had severe dementia. Severe dementia was defined with an Abbreviated Mental Test score of ≤ 6 [34, 35]. In Yangon, the surveyors visited 1,083 older adults and 610 were at home. Ten were excluded as they refused to participate in the survey (*n* = 6), had severe dementia or were bedridden (*n* = 4); the response rate was 98.4% in Yangon. In Bago, surveyors visited 1,044 older adults and 694 were at home. A total of 94 older adults were excluded as they had severe dementia or were bedridden; thus, the response rate was 86.5% in Bago. In total, 600 people each from the Yangon (222 men and 378 women) and Bago regions (261 men and 339 women) were surveyed.

### Questionnaire

A 14-page structured questionnaire—based on a questionnaire used in the "Japan Gerontological Evaluation Study" (JAGES) [36]—was developed for face-to-face interviews. JAGES was

established in 2010 as a nationwide, population-based prospective cohort study for older, community-dwelling, Japanese adults. The linguistic translation and validation process followed the "Linguistic Validation Manual for Health Outcome Assessments" [37]. The questionnaire was developed in English, translated into Burmese, and back-translated into English, to ensure clarity and consistency.

We hired research staff from the Myanma Perfect Research Company, a group with considerable experience in conducting epidemiological surveys in Myanmar. Before the commencement of the actual survey, a two-day training course on the research protocol, administration of the questionnaire, and ethical concerns was conducted for the interviewers.

A pilot study was carried out before the actual survey for face validity in the Urban Health Center of the Dagon township in Yangon. Participants were older adults aged 60 years or older who came to the center's out-patient clinic. We recruited 25 respondents who provided consent to participate in the pilot study, in June 2018. During the pilot study, the interviewers ensured sequence, flow, and clarity of the questionnaire. After the feedback from the interviewers, the questionnaire was revised accordingly. To avoid the question order bias, we positioned questions about depressive symptoms away from questions about subjective socioeconomic status (see S1 and S2 Questionnaires).

## Dependent variable

We assessed depressive symptoms using the 15-item version of the Geriatric Depression Scale (GDS), which was validated previously in other countries including Asian countries [38–41]. The GDS involves a simple yes/no format (see Q17 1)-15) in S1 and S2 Questionnaires), such that is easy to administer and score [42, 43]. Participants were classified into two groups: those exhibiting no depressive symptom (GDS score < 5), and those exhibiting depressive symptoms (GDS score ≥ 5) [39, 44–46].

## Independent variables

The wealth index, used as an indicator of objective SES, was calculated from household asset items using a method described in a previous report [47]. A principal component analysis was performed on the asset items (e.g., radio, black & white television, color television, Video/DVD player, electric fan, refrigerator, computer, store-bought furniture, personal music player, washing machine, gas cooker, electric cooker or rice cooker, air conditioner, bicycle, motorcycle, van/truck, microwave oven, mobile telephone, and internet). The principal component score was calculated based on the participants' possession of each item and used as the wealth index. Subjective SES was assessed by asking: "Which of the following best describes your current financial situation in light of general economic conditions?" The participants were asked to select from five options. Their perception of their current financial situation was: 1. very difficult, 2. difficult, 3. average, 4. comfortable, and 5. very comfortable. Based on their responses, participants were categorized as having "average or more" (answering 3, 4, or 5) or "difficult or very difficult" (answering 1 or 2) perceived SES, taking general economic conditions into consideration.

## Sociodemographic characteristics

The sociodemographic characteristics of the study participants included information regarding their residential area (Yangon or Bago), age, sex, illness during the preceding year, educational level (no school, the Buddhist monastic school, some/all primary school, middle/high school or higher), marital status (married or widowed/divorced/never married), living status (alone or not alone), religion (Buddhism or other), frequency of visits to religious facilities

(less than once per week, or once per week or more), and receipt of social support (giving and receiving emotional and instrumental help). Social support was assessed by asking four questions. The questions included: (1) Do you listen to someone else's concerns and complaints? (giving socioemotional support); (2) Do you take care of someone who is sick? (giving instrumental social support); (3) Do you have someone who listens to your concerns and complaints? (receiving emotional social support); and, (4) Do you have someone who takes care of you when you are sick? (receiving instrumental social support). For these questions, the possible responses were: 1. none; 2. spouse; 3. children living with them; 4. children or relatives living apart; 5. neighbor; 6. friend; and, 7. other. Based on their responses, participants were categorized as "having no social support" (i.e., answering with 'none') or "having social support" (answering with any of the choices between 2 and 7) [48].

## Statistical analyses

Sociodemographic characteristics were compared between participants who had depressive symptoms (GDS score $\geq$ 5) and those who did not have depressive symptoms (GDS score <5), using Pearson's chi-square test. A multivariable logistic regression analysis was performed to identify the factors associated with depressive symptoms. Variables for objective and subjective SES and the other variables with an associated p-value level less than 0.05 in bivariate analyses were simultaneously entered into a model. Adjusted odds ratios (AOR) were presented with 95% confidence intervals (CI). After performing an analysis on all the subjects, we also performed a stratified analysis by gender and region. We used STATA14 to perform all statistical analyses [49], and the statistical significance level was set at $p < .05$.

## Ethical considerations

The survey protocol was reviewed and approved by the ethical review committee of the Department of Medical Research at the Ministry of Health and Sports, the Republic of the Union of Myanmar, the World Health Organization ethics committee, the ethics board of the Niigata University, and the National Institute of Public Health in Japan. Written informed consent was obtained from all participants before the interviews. Voluntary participation and the right to withdraw participation at any time were assured. The study conformed to the principles of the Declaration of Helsinki.

## Results

### Characteristics of respondents

Among the 1,186 participants who answered GDS questions, 265 (22.3%) had depressive symptoms (GDS score $\geq$ 5) (Table 1). As for SES, 39.7% of participants had a low wealth index (objective SES) and 20.6% rated themselves as having a difficult/very difficult economic status (subjective SES). The rates of both low objective and subjective SES were significantly higher among respondents who had depressive symptoms (51.7% and 43.8%, respectively) than those who did not have depressive symptoms (36.3% and 13.9%, respectively).

Over half of the participants experienced illness during the preceding year (51.3%), and 8.6% received no school schooling. Both experiences of illness during the preceding year and no school were significantly higher among those who had depressive symptoms (64.9% and 10.6%, respectively), than those who did not have depressive symptoms (47.3% and 8.0%, respectively). Most of the participants did not live alone (94.4%), and the rate of participants who did not live alone was significantly higher among those who did not have depressive symptoms (96.0%) than those who had depressive symptoms (88.7%). Although most

**Table 1. Sociodemographic characteristics (N = 1,200).**

| | | | | GDS < 5 | | GDS > = 5 | | |
|---|---|---|---|---|---|---|---|---|
| | | n = 1,186 | % | n = 921 | % | n = 265 | % | p value |
| **Objective SES** | Middle/High | 714 | 60.2% | 586 | 63.6% | 128 | 48.3% | < .001 |
| **(Wealth index)** | Low | 471 | 39.7% | 334 | 36.3% | 137 | 51.7% | |
| | Missing | 1 | 0.1% | 1 | 0.1% | 0 | 0.0% | |
| **Subjective SES** | Average or more | 942 | 79.4% | 793 | 86.1% | 149 | 56.2% | < .001 |
| **(self-rated economic status)** | Difficult/Very difficult | 244 | 20.6% | 128 | 13.9% | 116 | 43.8% | |
| **Age** | 60–69 | 664 | 56.0% | 520 | 56.5% | 144 | 54.3% | 0.799 |
| | 70–79 | 376 | 31.7% | 290 | 31.5% | 86 | 32.5% | |
| | 80+ | 150 | 12.6% | 74 | 8.0% | 76 | 28.7% | |
| **Sex** | Male | 480 | 40.5% | 398 | 43.2% | 82 | 30.9% | < .001 |
| | Female | 706 | 59.5% | 523 | 56.8% | 183 | 69.1% | |
| **Illness during preceding year** | No | 575 | 48.5% | 482 | 52.3% | 93 | 35.1% | < .001 |
| | Yes | 608 | 51.3% | 436 | 47.3% | 172 | 64.9% | |
| | Missing | 3 | 0.3% | 3 | 0.3% | 0 | 0.0% | |
| **Education** | No school | 102 | 8.6% | 74 | 8.0% | 28 | 10.6% | < .001 |
| | Monastic school | 286 | 24.1% | 207 | 22.5% | 79 | 29.8% | |
| | Some/Finished primary | 415 | 35.0% | 308 | 33.4% | 107 | 40.4% | |
| | Middle school or higher | 383 | 32.3% | 332 | 36.0% | 51 | 19.2% | |
| **Region** | Yangon | 594 | 50.1% | 498 | 54.1% | 96 | 36.2% | < .001 |
| | Bago | 592 | 49.9% | 423 | 45.9% | 169 | 63.8% | |
| **Marital status** | Married | 638 | 53.8% | 515 | 55.9% | 123 | 46.4% | 0.006 |
| | Widow/Divorced/Never | 548 | 46.2% | 406 | 44.1% | 142 | 53.6% | |
| **Living status** | Alone | 67 | 5.6% | 37 | 4.0% | 30 | 11.3% | < .001 |
| | Not alone | 1119 | 94.4% | 884 | 96.0% | 235 | 88.7% | |
| **Social Support** | | | | | | | | |
| **Receiving emotional support** | No | 168 | 14.2% | 131 | 14.2% | 37 | 14.0% | 0.914 |
| | Yes | 1018 | 85.8% | 790 | 85.8% | 228 | 86.0% | |
| **Providing emotional support** | No | 188 | 15.9% | 138 | 15.0% | 50 | 18.9% | 0.127 |
| | Yes | 998 | 84.1% | 783 | 85.0% | 215 | 81.1% | |
| **Receiving instrumental support** | No | 26 | 2.2% | 13 | 1.4% | 13 | 4.9% | 0.001 |
| | Yes | 1160 | 97.8% | 908 | 98.6% | 252 | 95.1% | |
| **Providing instrumental support** | No | 260 | 21.9% | 200 | 21.7% | 60 | 22.6% | 0.748 |
| | Yes | 926 | 78.1% | 721 | 78.3% | 205 | 77.4% | |
| **Religion** | Buddhism | 1135 | 95.7% | 882 | 95.8% | 253 | 95.5% | 0.835 |
| | Other | 51 | 4.3% | 39 | 4.2% | 12 | 4.5% | |
| **Frequency of religious visits** | Less than once per week | 607 | 51.2% | 454 | 49.3% | 153 | 57.7% | 0.015 |
| | Once per week or more | 579 | 48.8% | 467 | 50.7% | 112 | 42.3% | |

p-value for chi-square test, GDS = Geriatric Depression Scale, SES = Socioeconomic status.

Number of participants who did not answer GDS questions = 14.

participants had social support (giving and receiving emotional and instrumental help), the rate of instrumental support received was significantly lower among those who had depressive symptoms compared with those who did not have depressive symptoms (95.1% and 98.6%, respectively). Nearly half of the respondents visited religious facilities once a week or more (48.8%), and the rate was significantly higher among those who did not have depressive symptoms than those who had depressive symptoms (50.7% and 42.3%, respectively).

## Associations between objective/subjective SES and depressive symptoms

Depressive symptoms were positively associated with being female (AOR: 1.64, 95% CI: 1.15–2.34), experiencing illness during the preceding year (AOR: 1.92, 95% CI: 1.41–2.61), and living in Bago (AOR: 1.62, 95% CI: 1.10–2.38). Depressive symptoms were negatively associated with receiving instrumental support (AOR: 0.31, 95% CI: 0.12–0.77) and frequency of visits to religious facilities once per week or more (AOR: 0.57, 95% CI: 0.42–0.77). Low subjective SES was positively associated with depressive symptoms (AOR: 4.18, 95% CI: 2.98–5.87) (Table 2). Low subjective SES was still significantly associated with depressive symptoms after being stratified by region—Yangon (AOR: 2.10, 95% CI: 1.08–4.05) and Bago (AOR: 5.65, 95% CI: 3.69–8.64) (Table 3).

Among the men in Yangon, receiving instrumental support and the frequency of visits to religious facilities once per week or more were both negatively associated with depressive symptoms (receiving instrumental support: AOR: 0.05, 95% CI: 0.01–0.38; frequency of visits to religious facilities: AOR: 0.23, 95% CI: 0.05–0.96) (Table 3).

Among the residents of Bago, low subjective SES was positively associated with depressive symptoms among both men and women (men: AOR: 8.97, 95% CI: 4.46–18.07; women: AOR: 4.45, 95% CI: 2.57–7.72). Meanwhile, the frequency of visits to religious facilities once per week or more was negatively associated with depressive symptoms (rural men: AOR: 0.45, 95% CI: 0.23–0.87; rural women: AOR: 0.39, 95% CI: 0.23–0.67) (Table 4). Variables that were not significantly associated with depressive symptoms can be found in Tables 2, 3 and 4.

**Table 2. Multivariate adjusted association between depressive symptoms and objective/subjective socioeconomic status among the older adults the two regions in Myanmar.**

| n = 1,182 | | AOR | 95% CI | |
|---|---|---|---|---|
| **Objective SES** | Middle/High | 1.00 | | |
| **(Wealth index)** | Low | 0.90 | 0.61 | 1.32 |
| **Subjective SES** | Average or more | 1.00 | | |
| **(self-rated economic status)** | Difficult/Very difficult | 4.18 | 2.98 | 5.87 |
| **Sex** | Male | 1.00 | | |
| | female | 1.64 | 1.15 | 2.34 |
| **Illness during preceding year** | No | 1.00 | | |
| | Yes | 1.92 | 1.42 | 2.61 |
| **Education** | No school | 1.00 | | |
| | The Buddhist monastic school | 1.42 | 0.80 | 2.51 |
| | Some/Finished primary | 1.46 | 0.84 | 2.53 |
| | Middle school or higher | 0.93 | 0.50 | 1.74 |
| **Region** | Yangon | 1.00 | | |
| | Bago | 1.62 | 1.10 | 2.38 |
| **Marital status** | Married | 1.00 | | |
| | Widow/Divorced/Never | 1.08 | 0.77 | 1.52 |
| **Living status** | Alone | 1.00 | | |
| | Not alone | 0.58 | 0.31 | 1.06 |
| **Receiving instrumental social support** | No | 1.00 | | |
| | Yes | 0.31 | 0.12 | 0.77 |
| **Frequency of visits to religious facilities** | Less than once per week | 1.00 | | |
| | Once per week or more | 0.57 | 0.42 | 0.77 |
| R2 = 0.14 | | | | |

AOR: Adjusted Odds Ratio, CI: Confidence Interval, SES: Socioeconomic Status.

**Table 3. Multivariate adjusted association between depressive symptoms and objective/subjective socioeconomic status among the urban (Yangon) and rural (Bago) older adults in Myanmar.**

| | n = 591 | AOR | 95%CI | | n = 591 | AOR | 95%CI | |
|---|---|---|---|---|---|---|---|---|
| | Urban (Yangon) | | | | Rural (Bago) | | | |
| **Objective SES** | Middle/High | 1.00 | | | | 1.00 | | |
| **(Wealth index)** | Low | 1.65 | 0.80 | 3.41 | | 0.74 | 0.47 | 1.17 |
| **Subjective SES** | Average or more | 1.00 | | | | 1.00 | | |
| **(self-rated economic status)** | Difficult/Very difficult | 2.10 | 1.08 | 4.05 | | 5.65 | 3.69 | 8.64 |
| **Sex** | Male | 1.00 | | | | 1.00 | | |
| | female | 2.96 | 1.57 | 5.58 | | 1.25 | 0.77 | 2.01 |
| **Illness during preceding year** | No | 1.00 | | | | 1.00 | | |
| | Yes | 1.99 | 1.22 | 3.25 | | 2.04 | 1.36 | 3.07 |
| **Education** | No school | 1.00 | | | | 1.00 | | |
| | The Buddhist monastic school | 1.50 | 0.54 | 4.14 | | 1.24 | 0.60 | 2.53 |
| | Some/Finished primary | 1.64 | 0.65 | 4.13 | | 1.16 | 0.57 | 2.34 |
| | Middle school or higher | 1.15 | 0.45 | 2.95 | | 0.75 | 0.29 | 1.90 |
| **Marital status** | Married | 1.00 | | | | 1.00 | | |
| | Widow/Divorced/Never | 1.76 | 1.06 | 2.93 | | 0.75 | 0.46 | 1.22 |
| **Living status** | Alone | 1.00 | | | | 1.00 | | |
| | Not alone | 0.73 | 0.24 | 2.21 | | 0.48 | 0.22 | 1.04 |
| **Receiving instrumental social support** | No | 1.00 | | | | 1.00 | | |
| | Yes | 0.10 | 0.02 | 0.40 | | 0.87 | 0.22 | 3.44 |
| **Frequency of visits to religious facilities** | Less than once per week | 1.00 | | | | 1.00 | | |
| | Once per week or more | 0.85 | 0.53 | 1.37 | | 0.42 | 0.28 | 0.63 |
| | R2 = 0.12 | | | | R2 = 0.17 | | | |

AOR: Adjusted Odds Ratio, CI: Confidence Interval, SES: Socioeconomic Status

## Discussion

The main contribution of this study was the identification of the associations between depressive symptoms and objective and subjective SES in older adults living in low-income settings in Myanmar, where the aging trend in population is expected to increase rapidly. A total of 22.1% of older adults had depressive symptoms.

Previous studies in middle- or high-income countries only observed a significant association between objective SES and mental health [5, 6, 9]. In our study, however, those with low subjective SES were more likely to experience depressive symptoms than those with average or higher subjective SES, even after adjusting for objective SES and other covariates.

Although a previous meta-analysis in middle- or high-income countries did not investigate the association between SES and mental health, it showed that subjective SES affects physical health more than objective SES [19]. To the best of our knowledge, this is the first study showing that subjective SES is also associated with mental health more than objective SES. Moreover, we found that low subjective SES was associated with depressive symptoms in rural older adults, but not in urban older adults.

Previous research on community-dwelling older adults in Myanmar indicated that 22.2% of them had depressive symptoms [26]. A large-scale survey on older adults conducted in Myanmar, also revealed that approximately 16% to 56% of them had depressive symptoms [27]; 22.3% of the present study falls within this category. The median prevalence rate of depressive symptoms for adults aged 60 years and above in the world is estimated to be 10.3%

**Table 4. Multivariate adjusted association between depressive symptoms and objective/subjective socioeconomic status among the male and female older adults in urban (Yangon) and rural areas (Bago) in Myanmar.**

| | | n = 219 | AOR | 95% CI | | n = 371 | AOR | 95% CI | | n = 255 | AOR | 95% CI | | n = 332 | AOR | 95% CI | |
|---|---|---|---|---|---|---|---|---|---|---|---|---|---|---|---|---|---|
| | | **Urban (Yangon) men** | | | | **Urban (Yangon) women** | | | | **Rural (Bago) men** | | | | **Rural (Bago) women** | | | |
| **Objective SES** | Middle/High | | 1.00 | | | | 1.00 | | | | 1.00 | | | | 1.00 | | |
| **(Wealth index)** | Low | | 1.27 | 0.22 | 7.52 | | 1.88 | 0.82 | 4.27 | | 0.77 | 0.37 | 1.58 | | 0.77 | 0.42 | 1.40 |
| **Subjective SES** | Average or more | | 1.00 | | | | 1.00 | | | | 1.00 | | | | 1.00 | | |
| **(self-rated economic status)** | Difficult/Very difficult | | 2.84 | 0.74 | 10.86 | | 1.91 | 0.88 | 4.14 | | 8.97 | 4.46 | 18.07 | | 4.45 | 2.57 | 7.72 |
| **Illness during preceding year** | No | | 1.00 | | | | 1.00 | | | | 1.00 | | | | 1.00 | | |
| | Yes | | 0.88 | 0.29 | 2.65 | | 2.51 | 1.42 | 4.44 | | 1.32 | 0.68 | 2.56 | | 2.59 | 1.50 | 4.46 |
| **Education** | No school | | 1.00 | | | | 1.00 | | | | 1.00 | | | | 1.00 | | |
| | The Buddhist monastic school | | 0.58 | 0.09 | 3.74 | | 1.66 | 0.57 | 4.80 | | 1.45 | 0.55 | 3.82 | | 1.24 | 0.58 | 2.65 |
| | Some/Finished primary | | 2.62 | 0.77 | 8.96 | | 1.59 | 0.61 | 4.15 | | 1.59 | 0.62 | 4.06 | | 0.96 | 0.46 | 2.01 |
| | Middle school or higher | | 1.00 | (omitted) | | | 1.24 | 0.46 | 3.35 | | 1.00 | (omitted) | | | 0.69 | 0.19 | 2.51 |
| **Marital status** | Married | | 1.00 | | | | 1.00 | | | | 1.00 | | | | 1.00 | | |
| | Widow/Divorced/Never | | 1.36 | 0.41 | 4.51 | | 1.96 | 1.08 | 3.53 | | 1.08 | 0.41 | 2.84 | | 0.63 | 0.36 | 1.12 |
| **Living status** | Alone | | 1.00 | | | | 1.00 | | | | 1.00 | | | | 1.00 | | |
| | Not alone | | 4.60 | 0.22 | 96.40 | | 0.48 | 0.14 | 1.66 | | 0.25 | 0.04 | 1.72 | | 0.53 | 0.22 | 1.25 |
| **Receiving instrumental social support** | No | | 1.00 | | | | 1.00 | | | | 1.00 | | | | 1.00 | | |
| | Yes | | 0.05 | 0.01 | 0.38 | | 0.12 | 0.01 | 1.19 | | 3.73 | 0.14 | 97.32 | | 0.59 | 0.12 | 2.96 |
| **Frequency of visits to religious facilities** | Less than once per week | | 1.00 | | | | 1.00 | | | | 1.00 | | | | 1.00 | | |
| | Once per week or more | | 0.23 | 0.05 | 0.96 | | 1.02 | 0.60 | 1.76 | | 0.45 | 0.23 | 0.87 | | 0.39 | 0.23 | 0.67 |
| | | R2 = 0.14 | | | | R2 = 0.10 | | | | R2 = 0.20 | | | | R2 = 0.15 | | | |

AOR: Adjusted Odds Ratio, CI: Confidence Interval, SES: Socioeconomic Status.

[50]. Although we cannot directly compare the prevalence in this study with that of previous studies because of differences in the study period, the measure of depressive symptoms used, and the inclusion and exclusion criteria for the study population, the prevalence in Myanmar was relatively higher than the global prevalence [50].

Although both low objective and subjective SES were significantly associated with depressive symptoms by a bivariate analysis, only subjective SES was associated after adjusting for objective SES and other covariates in the multiple regression model. Similar to the other covariates, sex and physical illness were associated with depressive symptoms in this study. It is well known that women are generally more likely to be depressed than men [51] and that physical illness is associated with depressive symptoms [52]. The present study reflected these findings. However, the AOR of low subjective SES compared with middle/high subjective SES was greater than that of the AORs of sex and physical illness.

This may be due to several factors. One reason may be that health disparities due to differences in subjective SES may increase depressive symptoms in older adults in Myanmar. Persistent inequalities exist in health outcomes in Myanmar's seven states and seven regions [53, 54]. According to Zaw et al. [54], conventional budget allocations related to population and infrastructure provide disproportionately more resources to regions with better health and less resources than to areas with high health needs in Myanmar. Even in Japan, considered an

egalitarian society—as reflected by a Gini coefficient of 63% in 2019 [55]—with relatively few inequalities in health [56], substantial social inequalities in mental health, measured by SSS, were identified [11]. In a previous systematic review and meta-analysis, there also appeared to be a consistent and statistically significant increase in the odds of coronary artery disease, hypertension, diabetes, and dyslipidemia when comparing low and high SSS [13]. Although we adjusted for illness during the preceding year, low subjective SES may affect physical health due to the perception of status differentiation, which could lead to the risk of having depressive symptoms.

Another reason may be related to negative psychological consequences through stress-related psychological pathways due to low subjective SES. The idea is supported by empirical evidence showing that low SSS is associated with higher physiological stress markers [57–59]. Evidence also suggests that the stress-related dysregulation of the hypothalamic–pituitary–adrenal (HPA) axis, a part of the neuroendocrine system controlling responses to stress, predicts the onset and recurrence of depression [60, 61]. From these perspectives, the neuroendocrine pathway may link low subjective SES to depressive symptoms.

An alternative explanation of why low subjective SES is associated with depressive symptoms may be related to the difference between the extent to which the subjective SES and objective SES can be captured. One study indicates that SSS might not only represent a cognitive average of current socioeconomic circumstances, but also take account of past trajectories and perceived future prospects [62]. Additionally, when people rank themselves on the SSS ladder, they might refer to socioeconomic factors other than (or additional to) objective SES [61, 63]. Similar to the SSS ladder, subjective SES may be a comprehensive measure of people's socioeconomic situation, beyond the traditional objective indicators of SES.

Our study also revealed that the association between subjective SES and depressive symptoms is particularly strong among rural participants. This could be caused by the impact of poverty in rural areas. According to the Poverty Report in Myanmar, rural inhabitants were 2.7 times more likely than urban inhabitants to be poor [64]. This may increase the risk of depressive symptoms in rural older adults. It was also found that higher frequency of visits to religious facilities (once a week or more) were negatively associated with depressive symptoms among both male and female older adults in the rural area. The finding is consistent with those of previous studies indicating that religiousness tends to be experienced and expressed strongly by older adults [65, 66] and people living in rural areas [67, 68]. There is a possibility that religiousness mediates the relationship between subjective SES and depressive symptoms for rural older adults in Myanmar, who may have been at a much greater risk of developing depressive symptoms had they not been religious.

This study has several limitations. First, the nature of the face-to-face interviews did not allow for the objective assessment of participants' situations [69]. The assessment may have caused social desirability bias, resulting in misreporting of depressive symptoms. Second, our measurement of depressive symptoms was based only on the GDS scores, without corroborating clinical evaluation may not be very accurate. However, the GDS is a validated instrument for assessing depressive symptoms and is used widely [38, 40, 41, 48, 70]. Third, it is unknown whether these findings are generalizable beyond the Yangon and Bago regions of Myanmar. Myanmar is composed of seven regions and seven states. Therefore, it is difficult to generalize the study findings to the population in Myanmar. However, we may be able to estimate situations of older adults in other regions by the level of urbanization of the selected regions. Ideally, this survey should be extended to include all surrounding regions in the future. Fourth, reverse causality could have occurred because of the nature of the cross-sectional design. Longitudinal studies, in which the cause-and-effect pathway is more reliable, are required to resolve this issue. Fifth, there was a large number of statistical tests for the sample size, and

there were wide confidence intervals, after stratification by sex and region. Therefore, there is a potential risk of false positives from multiple testing and decreasing accuracy after the stratification. Sixth, indicators for bonding social capital, which are derived from relationships between similar persons such as with respect to sociodemographic and socioeconomic characteristics, were not included in our analysis. Previous research suggested that rural areas are richer than urban areas in bonding social capital [71, 72]. Although we adjusted for instrumental social support, which was associated with depressive symptoms in a bivariate analysis, further studies are needed to examine the association between types of social support and human interactions and depressive symptoms, in Myanmar. Finally, we developed the questionnaire in the dominant language based on the "Linguistic Validation Manual for Health Outcome Assessment [37]." However, it is necessary to improve the accuracy of the questionnaire to minimize such errors and obtain higher-quality results. Despite these limitations, this study found that subjective SES had a greater association to depressive symptoms than objective SES, for older adults in the urban and rural areas in Myanmar. In addition, for the external validity of our findings, we expect that the international recession that began in 2018 has not had a significant influence on our findings, since the Gross Domestic Product (GDP) in Myanmar was equivalent to US$ 76.17 billion in 2018 [73], and the distribution of household income also increased between 2000–2018: The proportion of the population in the middle-income group (household income of US$ 5,000–34,999) rose from about 1.2% in 2000 to 20.6% in 2018 [74].

In conclusion, the association between subjective SES and depressive symptoms have been greater than that of objective SES and depressive symptoms in the two urban and rural areas of Myanmar, especially in the rural area. Considering not only material wealth, subjective SES should be important for decreasing depression in older adults in the area. Intervention programs for depression in older adults, which include social protection, sustainable livelihood, and wealth creation, should also consider regional differences in the context of subjective SES, by reducing economic disparities between rural and urban areas and within communities. A detailed study should also be conducted on how unique factors of the cultural background such as religiosity affect the mental health of older men and women in Myanmar.

## Supporting information

**S1 Questionnaires. Registration sheet for research project, "Healthy ageing in Myanmar (Myanmar).**
(PDF)

**S2 Questionnaires. Registration sheet for research project, "Healthy ageing in Myanmar (English).**
(PDF)

## Acknowledgments

We would like to thank all the study participants, and express our gratitude to Professor Than Win Nyunt, of the Department of Geriatric Medicine, Yangon General Hospital, Yangon, Myanmar. We also thank the Infectious Diseases Research Centre of Niigata University members, particularly Professor Reiko Saito and Professor Hisami Watanabe. In addition, we thank Ms. Saw Thu Nander, Mr. Yi Mynt Kyaw, and Myanma Perfect Research team members, who were deeply involved in the project implementation to conduct the survey. We wish to express our gratitude to the Japan Gerontological Evaluation Study principle investigator, Professor Katsunori Kondo and its core members, Dr. Naoki Kondo, Dr. Jun Aida, Dr. Toshiyuki Ojima, and Dr. Masashige Saito, who provided insightful advice regarding the project. We

would also like to thank: Dr. Hiroshi Murayama of the University of Tokyo, who advised us on the survey on aging, based on his professional experience; Ms. Akiko Tomita and Ms. Naoko Ito from the Japan International Cooperation Agency, who supported the conduction of the survey; Ms. Tomoko Manabe, who provided excellent secretarial support during the entire study; and, Dr. Reiko Hayashi from the National Institute of Population and Social Security Research, Japan, for his advice.

## Author Contributions

**Conceptualization:** Yuri Sasaki, Yugo Shobugawa, Ikuma Nozaki, Daisuke Takagi, Yuki Chihara, Hla Hla Win.

**Data curation:** Yuri Sasaki, Yugo Shobugawa, Ikuma Nozaki, Kay Thi Lwin, Poe Ei Zin, Thae Zarchi Bo, Hla Hla Win.

**Formal analysis:** Yuri Sasaki, Daisuke Takagi.

**Funding acquisition:** Yuri Sasaki, Yugo Shobugawa, Tomofumi Sone.

**Investigation:** Yuri Sasaki, Yugo Shobugawa, Yuiko Nagamine, Masafumi Funato, Yuki Chihara, Kay Thi Lwin, Poe Ei Zin, Thae Zarchi Bo.

**Methodology:** Yuri Sasaki, Yugo Shobugawa, Ikuma Nozaki, Daisuke Takagi.

**Project administration:** Yugo Shobugawa, Ikuma Nozaki, Daisuke Takagi, Yuiko Nagamine, Masafumi Funato, Yuki Shirakura, Hla Hla Win.

**Resources:** Yugo Shobugawa, Kay Thi Lwin.

**Supervision:** Yuri Sasaki, Yugo Shobugawa, Ikuma Nozaki, Yuiko Nagamine, Masafumi Funato, Yuki Chihara, Yuki Shirakura, Tomofumi Sone, Hla Hla Win.

**Validation:** Daisuke Takagi, Yuiko Nagamine.

**Writing – original draft:** Yuri Sasaki.

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
