## [Decision Letter · Decision Letter 0]

14 Jul 2020

PONE-D-20-15915

Association between depressive symptoms and objective/subjective socioeconomic status among older people in Myanmar

PLOS ONE

Dear Dr. Sasaki,

Thank you for submitting your manuscript to PLOS ONE. After careful consideration, we feel that it has merit but does not fully meet PLOS ONE’s publication criteria as it currently stands. Therefore, we invite you to submit a revised version of the manuscript that addresses the points raised during the review process.

We look forward to receiving your revised manuscript.

Kind regards,

Siyan Yi, MD, MHSc, PhD

Academic Editor

PLOS ONE

Journal requirements;

3. Please state whether you validated the questionnaire prior to testing on study participants. Please provide details regarding the validation group within the methods section.

Reviewers' comments:

Reviewer's Responses to Questions

**Comments to the Author**

1. Is the manuscript technically sound, and do the data support the conclusions?

Reviewer #1: No

Reviewer #2: Yes

Reviewer #3: Yes

Reviewer #4: Partly

2. Has the statistical analysis been performed appropriately and rigorously? 

Reviewer #1: No

Reviewer #2: Yes

Reviewer #3: Yes

Reviewer #4: Yes

3. Have the authors made all data underlying the findings in their manuscript fully available?

Reviewer #1: No

Reviewer #2: Yes

Reviewer #3: Yes

Reviewer #4: Yes

4. Is the manuscript presented in an intelligible fashion and written in standard English?

Reviewer #1: No

Reviewer #2: No

Reviewer #3: No

Reviewer #4: Yes

5. Review Comments to the Author

Reviewer #1: Thank you for the opportunity to review this study into the associations between objective and subjective socioeconomic status and depression among older people in Myanmar. The study reported on the cross-sectional statistical analysis of data from a survey. The authors found statistically significant associations following regression adjustment between subjective but not objective socioeconomic status, the association was strong in rural areas and behaviours like religious participation seemed to alter the associations.

This is an important topic for research to help us understand the mechanisms through which socioeconomic inequalities lead to health impacts. However, I have a number of predominantly methodological concerns that need to be addressed before publication.

1. Line 24: ‘It influences mental health through adverse effects on physical surroundings and psychosocial experiences.’ This suggests a causal relationship, which I don’t think can be made. An action is being attributed to inequalities that are not animate, and even then the causal pathways are much more complicated than described. This confusion may result from translation from the authors own language to English?

2. Line 26: ‘However, no studies have been conducted on the impact of both SES on depressive symptoms in older people.’ This sentence is not particularly clear, it would be better to say ‘both objective and subjective SES’ to be clear. Moveover, a quick search of the Web of Science identified handful of papers already published on this topic, and therefore I don’t think the sentence is correct. The distinctive effects of objective and subjective socioeconomic status have been recognised for quite a few years and have been the focus of significant research. Depression might not been examined specifically in each of these papers, but as the authors note in the discussion their questionnaire would not necessarily distinguish diagnosed cases of depression, and therefore the outcome might be identifying life satisfaction as much as depression. Here are the DOI of some of the papers I found:

• 10.1016/j.archger.2016.07.006

• 10.1186/1471-2458-11-166

• 10.1016/j.socscimed.2008.03.038

• 10.1016/j.socscimed.2006.06.025

• 10.1111/j.1532-5415.2005.53169.x

Further development of the introduction and discussion are needed to represent this wider body of literature. Myanmar seems to be a novel study location, and therefore the low-income setting is more unusual, but great inclusion of the previous literature is needed.

3. Line 91: ‘In rural areas, some villages exist under the village tract.’ I think the sentence is meant to say that there can be multiple villages within a single village tract.

4. Line 96: ‘we considered Yangon as representative of urban areas and Bago as representative of rural areas.’ Additional information is needed in order to justify this sentence, the authors opinion is not sufficient. More broadly speaking more information is needed comparing the sample with the population to explore any possibly biases. As it stands we do not have any way of assessing whether the findings might just relate to a biased sample, or might relate to a wider problem across Myanmar?

5. Line 260: ‘studies should be needed as to what types of social supports and human interaction are’. The English in this sentence is not great, I think you could talk about further studies being needed…

6. It would be useful to know something about the order in which the questions were asked, as this could have influenced the responses received. If the depression questions led straight onto the subjective socioeconomic status questions, the participant’s mindset might have led to poorer responses.

7. It would be good to give some insight into whether the collection of data in 2018 might have impacted the findings. An international recession began in 2018, might this have impacted on findings?

8. As an epidemiological study I think the reporting needed to be improved to provide information about bias and aid the understanding of external validity. Using a reporting guideline like the STROBE checklist would ensure that all the relevant details are reported.

9. There are a large number of statistical tests undertaken for the sample size, given that models for urban/rural and gender are presented. Therefore, some discussion of the potential risk of false positives from the multiple testing is needed. Also, there is some evidence of very wide confidence intervals which suggests small numbers of participants in those groups. The assumptions of the models should be tests to ensure outliers are not overly impacting on the results. A sentence reporting that no breeches of assumptions were found, should be included in the paper.

10. The authors report that the study data are available upon request, which does not match their response that the data are fully available. I completely understand the restrictions around this sensitive data, but the journal gives the author space to explain that the data are only available in specific circumstances.

Overall, the study is interesting, but the paper needs to be improved in terms of links to existing literature and reporting of the methods. The findings in relation to religious practice are particularly interesting. If the research questions were more specific, the paper might be tightened up.

Reviewer #2: This paper is an interesting study on different contributions of objective and subjective socioeconomic status to depressive symptoms among Burmese older people. However, I have the following concerns.

The authors should briefly explain the situation in the mental health care system in Myanmar’s urban and rural areas in the introduction section. It is essential to understand “social support” available in the country at the time of this study. Information on the role of religious facilities in Myanmar’s daily life should also be added.

Lines 101–129

The authors need to reconstruct sections between Study tools and Confounding variables. All the study tools should be mentioned in the Study tools section.

What was first translated into English in line 103? “Japan Gerontological Evaluation Study” is the name of a tool? The authors also need to briefly explain what “abbreviated mental test” is like, including scoring and evaluation information.

When the authors use any evaluation tools in a different language for the very first time, it is essential to show the linguistic translation and validation process. The explanation in this manuscript is too vague. The authors should refer to Linguistic validation manual for health outcome assessments (2012, MAPI Institute) or other previous validation studies with the translation process on how to carry out translation and validation procedures to create an evaluation tool in a different language. The authors need to show this process in detail and mention the insufficiency, if any, in the limitation section. The same process needs to be applied to GDS and the wealth index as well.

Lines 225–227

The authors should refer to any previous articles or data on “which has been considered an egalitarian society with relatively few inequalities in health” to state that this comment is not just the authors’ personal belief.

Lines 238–

Bonding social capital was not directly questioned in this research. This should be mentioned in the limitation section.

Lines 273–274

The authors referred to the paper by Sommanustweechai et al. (2016)(45), but this previous study does not mention GDS validation. The papers by Yesavage et al. (1982) Sasaki et al. (2019) can not confirm linguistic translation and validation in Burmese language and Burmese cultural background.

Line 160 and other

There are some vague expressions such as “These were ...” in line 160.

Line 274

“Addition” should be “In addition” or “Additionally” in line 274.

Reviewer #3: Dear author,

Even though there has been published depressive symptoms among elder Myanmar population in both institutional and community dwelling, this paper is the first to examine the association between depressive symptoms and objective/ subjective socioeconomic status among elder Myanmar population. It addresses a significant gap in the literature. Thus, this paper has the potential to make a significant contribution. However, I have provided some suggestions to improve the manuscript below.

Abstract

In the abstract, in this statement “no studies have been conducted on the impact of both SES on depressive symptoms in older people”, please specify where is?

Introduction

1. Please elaborate on the contextual factors affecting depression and the status of socio-economic condition in elder Myanmar population.

2. More cogent and clear arguments need to be made as to why the associations were investigated between objective, subjective social economic status and depressive symptoms.

Methods:

1. The study design that you mentioned in line 85, 86 and 87 is rather vague. Please clearly specify your study design and time period.

2. Reason as to why Yangon and Bago region was chosen seem tenuous. There is also rural areas in Yangon region why do you choose Bago region as rural (area).

3. In your abstract, you described multi-stage sampling so you need to mentioned detail sampling procedure.

4. In line 93 “Differences between wards and village tracts involve the degree of urbanization” What does it mean?

5. Sample size and/or participation rate need to be mentioned

6. How did you train the trained surveyors and how did you recruit the participants?

7. Description of survey tools is rather vague, please clarify clearly. What is Japan Gerontological Evaluation Study and how did you used in your study.

8. You described the exclusion criteria in the study tool in “line 104”. You should mention both inclusion and exclusion criteria under the study design and participants.

9. In your dependent variable, why do you use GDS question as screening tool for symptoms of depression in elder population. Please specify short version of your GDS items and mentioned where this question validated (only in Japan or other countries).

10. In independent variable, please a brief specify wealth index indicators.

11. Regarding confounding variable, how to do you ask social support question. Please specify this question.

Discussion

1. While discussing prevalence of depressive symptoms, there has been recently published prevalence of depressive symptoms in elder Myanmar population and you should also use it as reference.

2. While appreciating your work, Have your screening instrument been standardized in Myanmar? If not, please talk about it in limitation.

3. More cogent and clear arguments need to be made as to why low subjective SES is associated with depressive symptoms in older adults.

4. I am not sure as to why some of the results were presented in the Discussion Section (line 211, 214, 215, 216)

5. I found the arguments presented in the Discussion section rather disjointed. There needs to be more flow of discussion that tells a clear story. Some of the arguments are lacking in substance and/or details- for example, in line 216 to 219.There needs to be a clear discussion.

Conclusion

Clarification and more meaningful detail/interpretation required.

Reviewer #4: This study is an examination of the relationship between objective socioeconomic status, subjective socioeconomic status and depressive symptoms among community-dwelling older adults in Myanmar. This study utilized multistage random sampling and face-to-face interviews to collect data on 1200 participants aged 60 years and above. Authors found that subjective SES was associated with greater depressive symptomology, and this association was particularly strong among those living in rural communities. This is an interesting study, yet lacks sufficient detail and clarity in the writing to be considered for publication. My comments and suggestions for improvement are below:

Abstract

1. Methods: I suggest including the number of participants in the first sentence of this section.

2. Methods: what is the “cutoff of five” referring to in the sentence regarding the Geriatric Depression Scale?

Introduction

1. What is meant by objective and subjective socioeconomic status? Authors need to provide definition to these concepts, as well as contrasts between the two concepts.

2. Line 63: Need to clarify how the perception of lower/higher SES is determined. Are individuals self-reporting? Are they asked to compare their status to others? Please expand the discussion of this concept.

3. Line 74: What is meant by the “socioeconomic cost of depression”? Please expand discussion.

4. Line 77: Needs clarity in writing. Authors should present the “doubling” statistic, or the 25% increase statistic. Both are not needed.

5. Line 80: Unclear what authors are examining in the study. Are you interested in both objective AND subjective SES? Examining in the same models? In separate models? Please be precise.

Methods

1. Line 85: Include how many participants are in the study up front.

2. Line 114: Please provide a more in depth explanation of the objective SES measure.

3. Line 115: Better describe the subjective SES measure. What does “average” mean in the possible responses? Average compared to who? What is “very difficult” or “very comfortable” supposed to mean?

4. Line 127: Please better describe the social support variable. What is emotional vs instrumental help? Are these different variables? Or are they all included into 1 variable?

5. Line 135: Were all variables included in 1 model? Was there a model for objective SES and a different model for subjective SES? This needs to be better described.

6. Line 136: When and how was statistical significance assessed for the covariates?

Results

1. Line 158: among those who did not what?

2. Line 162: among those who did not what?

3. Line 163: What do authors mean by “significantly higher”? Clarify this.

4. Line 166: Need to clarify what is meant by emotional vs instrumental support

5. Line 180: report AOR and CI for objective SES

6. Line 182: report AOR and CI for objective SES

7. Line 183: Authors just presented regional stratification. No need to mention here.

8. Line 185: Need clarity as to why the authors are only reporting results for the Bago region. This needs to be clarified in the methods section.

Discussion

1. Line 222: Authors need to describe how SSS compares to SES and why this is applicable.

2. Line 238: Authors need to better link their results to this discussion of the potentially unique aspects of rural life. Also, can authors look at the rural vs urban differences in social support and tie that into this section? Social support may be a good proxy for social capital. Further explore this variable in a post-hoc analysis and greatly expand this discussion.

Table 1.

1. What does the “Missing” column add to the table? This data can be noted elsewhere on the table and save space.

Table 2.

1. Add more space between the rows. This will make it easier to read.

2. Why is Monastic education tied in with No School? Should these be 2 different categories?

3. Why are widowed and divorced the same category? Could they have different effects on depression?

Tables 2a-2b

1. Add more space between the rows. This will make it easier to read.

2. Why is Monastic education tied in with No School? Should these be 2 different categories?

3. Why are widowed and divorced the same category? Could they have different effects on depression?

4. Include the rural and urban results on the same table, with the AOR and CI next to one another. This will make it easier to compare. I don’t need to see the SE or the z statistic.

Tables 2c-2f

1. Add more space between the rows. This will make it easier to read.

2. Why is Monastic education tied in with No School? Should these be 2 different categories?

3. Why are widowed and divorced the same category? Could they have different effects on depression?

4. Include the rural and urban results for men and women on the same table, with the AOR and CI next to one another. This will make it easier to compare. I don’t need to see the SE or the z statistic.

6. PLOS authors have the option to publish the peer review history of their article (what does this mean?). If published, this will include your full peer review and any attached files.

Reviewer #1: **Yes: **Andrew James Williams

Reviewer #2: No

Reviewer #3: No

Reviewer #4: No

---

## [Author Response · Author response to Decision Letter 0]

26 Sep 2020

Please see attached file "Response to Editors and Reviewers (R1)"

---

## [Decision Letter · Decision Letter 1]

29 Oct 2020

PONE-D-20-15915R1

Association between depressive symptoms and objective/subjective socioeconomic status among older adults of two regions in Myanmar

PLOS ONE

Dear Dr. Sasaki,

Thank you for submitting your manuscript to PLOS ONE. After careful consideration, we feel that it has merit but does not fully meet PLOS ONE’s publication criteria as it currently stands. Therefore, we invite you to submit a revised version of the manuscript that addresses the points raised during the review process.

We look forward to receiving your revised manuscript.

Kind regards,

Siyan Yi, MD, MHSc, PhD

Academic Editor

PLOS ONE

Additional Editor Comments:

We thank the authors for carefully addressing the reviewers' comments. The manuscript has been greatly improved. However, we still have received comments from two reviewers, mostly for improving the writing quality and readability. Please also take this opportunity to proofread your paper before re-submission. You may not have a chance to make any further changes should the paper be accepted by PLOS ONE.

Reviewers' comments:

Reviewer's Responses to Questions

**Comments to the Author**

1. If the authors have adequately addressed your comments raised in a previous round of review and you feel that this manuscript is now acceptable for publication, you may indicate that here to bypass the “Comments to the Author” section, enter your conflict of interest statement in the “Confidential to Editor” section, and submit your "Accept" recommendation.

Reviewer #1: (No Response)

Reviewer #2: All comments have been addressed

Reviewer #3: All comments have been addressed

Reviewer #4: All comments have been addressed

2. Is the manuscript technically sound, and do the data support the conclusions?

Reviewer #1: Yes

Reviewer #2: Yes

Reviewer #3: Yes

Reviewer #4: Yes

3. Has the statistical analysis been performed appropriately and rigorously? 

Reviewer #1: Yes

Reviewer #2: Yes

Reviewer #3: Yes

Reviewer #4: Yes

4. Have the authors made all data underlying the findings in their manuscript fully available?

Reviewer #1: Yes

Reviewer #2: Yes

Reviewer #3: Yes

Reviewer #4: Yes

5. Is the manuscript presented in an intelligible fashion and written in standard English?

Reviewer #1: No

Reviewer #2: No

Reviewer #3: Yes

Reviewer #4: Yes

6. Review Comments to the Author

Reviewer #1: I would like to thank the authors for the extensive revisions they have undertaken in response to all the reviewers comments. I think these have greatly improved the paper. There are a couple of remaining points I would like to raise, and otherwise my suggestions are related to improving the clarity of the language.

1. Having expanded the discussion of existing literature in the Introduction, it would be good to reflect back on this and demonstrate where the studies findings agree or disagree with that literature. Did you find anything novel, and are there new research questions that we now need to address?

2. Lines 331-333, more information is needed to support the statements made in this sentence. What is it about the resource allocation that does not align with reducing health disparities?

3. Thank you for the explanation about the lack of impact in Myanmar from the 2018 recession. I think it would be helpful to add a sentence in the discussion to explain that the 2018 recession had minimal effect on Myanmar, just to help anyone considering the external validity of your findings and to demonstrate that this had been considered.

Sentences were extra clarity is needed

• Lines 46-48

• Line 56, I wonder if describing objective SES using the term ‘in relation to others’ could be confusing.

• Line 77, I wonder if it should be ‘concepts’ plural rather than ‘concept’ singular.

• Lines 85-86

• Lines 93-96 could be broken into two sentences to be clearer.

• Line 98 needs more detail, explanation and a reference

• Line 102, it should be fewer activities of daily living. Is the point that people undertake fewer activities or that they struggle with them more?

• Lines 107-109 I think this sentence introduced a new concept that is not sufficiently explained, so the sentence can be deleted.

• Line 113, I am not sure of the meaning of ‘Myanmar’s ethic areas’, could this be phrased differently to make more sense to people outside of Myanmar?

• Line 148, 1,044 as you have used the comma separator elsewhere.

• Lines 169-170, I think ‘who’ needs to be added before came to make the meaning clearer.

• Line 232, the term multivariate implies that you were using models that predicted multiple dependent variables, where I think you want ‘multivariable’ which means that there were multiple independent variables being used to predict a single dependent variable.

• Line 234, add a citation for Stata

• Lines 285-286, I think it is best to day that they were not ‘significantly’ associated, as the lack of statistical significance does not imply the lack of association, just that you cannot accept the alternative hypothesis.

• Lines 307-308, the beginning of this sentence is not clear.

• Line 393, I would say you undertook a ‘large number of statistical tests for the sample size’.

• Line 417-418, why is it necessary?

Any additional steps that can be taken to improve the language throughout the paper would be helpful.

Reviewer #2: I agree with the authors’ response and correction to my previous review comments.

Still, the authors should review their logical flow and expressions.

For example, on page 2, the background section in the abstract, this paragraph starts from ‘This study examines....’ Here, the authors should start from this study’s background, then explain how the background made the authors launch this study. On page 4, the introduction section, ‘Research’ should be ‘Previous research,’ and it looks strange to find the word ‘conclusion’ in the very first sentence of the paper. There are numerous better alternatives such as ‘Previous studies demonstrated ...,’ ‘...has been recognized...’, and so on. Additionally, a description of SES was also embedded into one sentence, making the sentence a little unreadable. The authors should explain SES first, then refer to the relation between SES and poor health in older adults. There are other similar problems that require revision.

Reviewer #3: The author has provided response in detail to the comments from the previous review and has adequately addressed my major concerns.

Reviewer #4: (No Response)

7. PLOS authors have the option to publish the peer review history of their article (what does this mean?). If published, this will include your full peer review and any attached files.

Reviewer #1: **Yes: **Andrew James Williams

Reviewer #2: No

Reviewer #3: No

Reviewer #4: No

---

## [Author Response · Author response to Decision Letter 1]

9 Nov 2020

Response to reviewers

We wish to thank the reviewers for giving us helpful comments regarding our manuscript. We have revised our manuscript accordingly.

Reviewer comment 1-1: 

Having expanded the discussion of existing literature in the Introduction, it would be good to reflect back on this and demonstrate where the studies findings agree or disagree with that literature. Did you find anything novel, and are there new research questions that we now need to address?

Our reply 1-1

We thank the reviewer for this precious comment.

We have added the followings in Discussion

➡

The result was not in accordance with studies in middle- or high-income countries that only investigated the association between objective SES and mental health (5, 6, 9). However, it was similar to the results of a previous meta-analysis showing that subjective SES affects physical health more than objective SES (19). As far as we know, this is the first study showing similar results for mental health. Moreover, our study also found that, after being stratified by region and sex, low subjective SES was still associated with depressive symptoms in rural older adults, but not in urban older adults.

///

A detailed study should also be conducted on how unique factors of the cultural background such as religiosity affect the mental health of older men and women in each region in Myanmar.

Reviewer comment 1-2: 

Lines 331-333, more information is needed to support the statements made in this sentence. What is it about the resource allocation that does not align with reducing health disparities?

Our reply 1-2

Thank you for this important comment. We have added the followings

For example, according to Zaw et al. (52), conventional budget allocations related to population and infrastructure provide disproportionately more resources to regions with better health and less resources to some areas with high health needs.

Reviewer comment 1-3: 

Thank you for the explanation about the lack of impact in Myanmar from the 2018 recession. I think it would be helpful to add a sentence in the discussion to explain that the 2018 recession had minimal effect on Myanmar, just to help anyone considering the external validity of your findings and to demonstrate that this had been considered.

Our reply 1-3

Thank you for this comment. 

We have added the followings in Discussion section.

➡

In addition, for the external validity of our findings, we expect that the international recession that began in 2018 has not had a significant influence on our findings, since the Gross Domestic Product (GDP) in Myanmar was equivalent to US$ 76.17 billion in 2018 (80), and the distribution of household income also increased between 2000–2018: The proportion of the population in the middle-income group (household income of US$ 5,000–34,999) rose from about 1.2% in 2000 to 20.6% in 2018 (81).

Reviewer comment 1-4: 

Sentences were extra clarity is needed

Our reply 1-4. 

Thank you for carefully reviewing our manuscript.

We have revised them and had native speakers of English proofread our English writing, accordingly. 

We 

• Lines 46-48

Interventions for depression in older adults should consider regional differences in the context of their subjective SES by reducing disparities among the community in Myanmar.

➡

Interventions for depression in older adults should take regional differences into consideration in the context of subjective SES by reducing socio-economic disparities among the communities of Myanmar.

• Line 56, I wonder if describing objective SES using the term ‘in relation to others’ could be confusing.

Thank you for this comment. We have deleted ‘in relation to others’ as follows.

—defined as the economic and social position in relation to others, such as working status, household wealth, and poverty status (1-4)—is///

➡

Objective socioeconomic status (SES) is defined as the economic and social position such as working status, household wealth, and poverty status (1-4). 

• Line 77, I wonder if it should be ‘concepts’ plural rather than ‘concept’ singular.

Thank you for this comment. We have revised it based on your comment.

///which is defined as a person’s perceived rank relative to others in their group, and concept similar to subjective SES///

➡

///which is defined as a person’s perceived rank relative to others in their group, and concepts similar to subjective SES///

• Lines 85-86

Therefore, both objective and subjective SES are hypothesized to be one of the most significant disparities, where physical and psychological health are poor among individuals of lower SES.

➡

Therefore, both objective and subjective SES are hypothesized to indicate the most significant disparities, such that individuals of lower SES tend to have physically and psychologically poor health.

• Lines 93-96 could be broken into two sentences to be clearer.

Thank you for this comment. We have revised it as follows.

However, depressive symptoms are common among older adults as the aging population grows, especially in Asian countries (21), and its socioeconomic cost is vast due to higher rates of morbidity and mortality and increased healthcare utilization and economic cost compared to younger adults (22).

➡

However, depressive symptoms are common among older adults as the aging population grows, especially in Asian countries (21).The socioeconomic cost for older adults is vast due to higher rates of morbidity and mortality and increased healthcare utilization and economic cost compared to younger adults (22).

• Line 98 needs more detail, explanation and a reference

Thank you for this comment. We have described it in detail and added references.

The status of depression is also expected to be severe. ➡

The status of depression is also expected to be severe (24). Depression and anxiety account for five per cent of disability-adjusted life years, which puts them in the top 10 contributors to disability in Myanmar (24, 25). However, there is no policy on mental health in Myanmar. As a result, mental health services have not received priority in primary healthcare, preventing thousands of people from accessing the mental health services they need (24).

• Line 102, it should be fewer activities of daily living. Is the point that people undertake fewer activities or that they struggle with them more?

Thank you for this comment. We described it clearly as follows. 

Studies in Myanmar have found that depressive symptoms in older adults were strongly associated with less activities of daily living and lower quality of life (26) and economic and health status (27).

➡

Studies in Myanmar have found that depressive symptoms in older adults were strongly associated with less independence in seven items (walking, ascending and descending stairs, feeding, dressing, going to the toilet, bathing, and grooming) and lower quality of life (26) and economic and health status (27)

• Lines 107-109 I think this sentence introduced a new concept that is not sufficiently explained, so the sentence can be deleted.

Thank you for this comment. We have deleted the sentence below.

Certain studies focusing on the effects of Vipassana mediation indicated that participants in the mediation courses reported reductions in anxiety and stress (29, 30).

• Line 113, I am not sure of the meaning of ‘Myanmar’s ethic areas’, could this be phrased differently to make more sense to people outside of Myanmar?

Thank you for this question. We have intended to refer to the internal conflicts. We have revised the sentence as follows.

///and ongoing conflicts in Myanmar’s ethnic areas (30-32).

➡

/// and the ongoing internal conflicts in some states of Myanmar (30-32).

• Line 148, 1,044 as you have used the comma separator elsewhere.

Thank you for this comment. We have revised it.

In Bago, surveyors visited 1044 older adults///

➡

In Bago, surveyors visited 1,044 older adults///

• Lines 169-170, I think ‘who’ needs to be added before came to make the meaning clearer.

Thank you for this comment. We have added ‘who’. 

Participants were the older adults, age of above 60 came to the out-patient clinic in the center.

➡

Participants were the older adults, age of above 60 who came to the out-patient clinic in the center.

• Line 232, the term multivariate implies that you were using models that predicted multiple dependent variables, where I think you want ‘multivariable’ which means that there were multiple independent variables being used to predict a single dependent variable.

Thank you for this comment. We have revised it as follows.

Variables for objective and subjective SES and the other variables with an associated p-value level less than 0.05 in the bivariate analysis were entered into the same model using multiple logistic regression.

➡

Variables for objective and subjective SES and the other variables with an associated p-value level less than 0.05 in the bivariate analysis were entered into the same model using multivariable logistic regression. The multivariable adjusted results were expressed as adjusted odds ratios (AOR) with 95% confidence intervals (CI).

• Line 234, add a citation for Stata

Thank you for this comment. We have added a citation. 

We used STATA14 to perform all statistical analyses

➡

We used STATA14 to perform all statistical analyses (48)

• Lines 285-286, I think it is best to say that they were not ‘significantly’ associated, as the lack of statistical significance does not imply the lack of association, just that you cannot accept the alternative hypothesis.

Thank you for this comment. We have revised it as follows.

Among men and women in Yangon, both low objective and subjective SES were not associated with depressive symptoms

➡

Among men and women in Yangon, both low objective and subjective SES were not significantly associated with depressive symptoms

• Lines 307-308, the beginning of this sentence is not clear.

Thank you for this comment. We have revised the sentence as follows.

The model with potential confounding factors was adjusted, suggesting that older adults with low subjective SES were more likely to experience depressive symptoms relative to those with average or more subjective SES, ///

➡

Those with low subjective SES were more likely to experience depressive symptoms than those with average or higher subjective SES, even after adjusting for potential confounding factors.

• Line 393, I would say you undertook a ‘large number of statistical tests for the sample size’.

Thank you for this comment. We have revised it as follows. 

Fourth, there were several statistical tests undertaken for the sample size///

➡

Fourth, there was a large number of statistical tests for the sample size ///

• Line 417-418, why is it necessary？

Thank you for this comment. We have added the reasons as follows

However, it is necessary to improve the accuracy of the questionnaire.

➡

However, it is necessary to improve the accuracy of the questionnaire to minimize such errors and obtain higher-quality results.

Reviewer #2: I agree with the authors’ response and correction to my previous review comments.

Still, the authors should review their logical flow and expressions.

Reviewer comment 2-1 

On page 2, the background section in the abstract, this paragraph starts from ‘This study examines....’ Here, the authors should start from this study’s background, then explain how the background made the authors launch this study. 

Our reply 2-1. 

Thank you for this valuable comment. 

We have revised it as follows. 

➡

Background

Low objective SES (e.g., household wealth; working and poverty status) has been correlated with poor physical and mental health. Some suggest that subjective SES (perceptions of having lower SES) is also as important. However, no studies have been conducted on the impact of both objective and subjective SES on depressive symptoms in older adults in Myanmar. This study examines whether subjective or objective socioeconomic status (SES) is associated with depressive symptoms in older adults in Myanmar.

Reviewer comment 2-2

On page 4, the introduction section, ‘Research’ should be ‘Previous research,’ and it looks strange to find the word ‘conclusion’ in the very first sentence of the paper. There are numerous better alternatives such as ‘Previous studies demonstrated ...,’ ‘...has been recognized...’, and so on.

A description of SES was also embedded into one sentence, making the sentence a little unreadable. The authors should explain SES first, then refer to the relation between SES and poor health in older adults.

Our reply 2-2. 

Thank you for this important comment.

We have revised it based on your comment.

➡

Objective socioeconomic status (SES) is defined as the economic and social position such as working status, household wealth, and poverty status (1-4). Previous studies have been recognized that low objective SES is correlated with poor health in older adults (4-8).///

Subjective SES is defined as a person’s conception of his or her position compared with that of others (1-3) (e.g., perceptions of having a lower/higher SES, which is determined by whether respondents rated themselves as lower than middle class in the country or the community; lower ratings suggested a lower SES) (11-13).

---

## [Editor Report · Decision Letter 2]

20 Nov 2020

PONE-D-20-15915R2

Association between depressive symptoms and objective/subjective socioeconomic status among older adults of two regions in Myanmar

PLOS ONE

Dear Dr. Sasaki,

Thank you for submitting your manuscript to PLOS ONE. After careful consideration, we feel that it has merit but does not fully meet PLOS ONE’s publication criteria as it currently stands. Therefore, we invite you to submit a revised version of the manuscript that addresses the points raised during the review process.

We look forward to receiving your revised manuscript.

Kind regards,

Siyan Yi, MD, MHSc, PhD

Academic Editor

PLOS ONE

Additional Editor Comments (if provided):

Thank you for your efforts in addressing our reviewers’ comments. The manuscript has been remarkably improved. However, before this manuscript can be published, the writing quality requires substantial improvement to ensure the manuscript's readability and accuracy. In the attached file, you can find examples of incorrect sentences, long sentences, excessive paragraphs, grammatical errors, and typos that can be easily found throughout the text and need corrections. Please note that my comments are exhaustive. Please proofread the revised manuscript carefully before resubmission.

---

## [Author Response · Author response to Decision Letter 2]

30 Dec 2020

Response to Reviewers

Re: PONE-D-20-15915(R2)

Association between depressive symptoms and objective/subjective socioeconomic status among older adults of two regions in Myanmar

Response to the editors

We wish to thank the editors for giving us helpful and precious comments regarding our manuscript. We have revised our manuscript accordingly.

Editor comment 1: 

Abstract

Please ensure that the manuscript's style and formats are aligned with those of PLOS ONE, including spacing, capitalization, punctuations, etc.

Our reply 1:

We have ensured that the manuscript's style and format are aligned with those of PLOS ONE (https://journals.plos.org/plosone/s/submission-guidelines).

Editor comment 2: 

Please ensure that all abbreviations/acronyms are spelled out at the first use (e.g., SES, line 21). The abbreviations should be used consistently afterward (e.g., SES, line 26).

Our reply 2:

We have revised it as follows:

Low objective socioeconomic status (SES) has been correlated with poor physical and mental health among older adults. Some studies suggest that subjective SES is also important for ensuring sound physical and mental health among older adults. However, few studies have been conducted on the impact of both objective and subjective SES on mental health among older adults. This study examines whether objective or subjective SES is associated with depressive symptoms in older adults in Myanmar.

Editor comment 3: 

Please use a comma, instead of semi-column, between words or phrases (e.g., line 21). This comment is applied throughout the manuscript.

Our reply 3:

We have taken into account your advice and adjusted for the same throughout the manuscript. Additionally, we have used the semicolon only to separate long phrases to avoid the grammatical error of comma splice and to distinguish between data of two or more categories. Kindly check our adjustments throughout the manuscript. 

Editor comment 4: 

Please ensure consistencies, e.g., socioeconomic (line 26) or socio-economic (line 49).

Our reply 4:

We have consistently used "socioeconomic" throughout the manuscript.

Editor comment 5: 

Background:

- First sentence: … has been correlated with poor physical and mental health. Among whom? Please state the populations specifically.

Our reply 5:

We have specified the population as follows.

Low objective socioeconomic status (SES) has been correlated with poor physical and mental health among older adults.

Editor comment 6: 

Background:

- Second sentence: Some suggest that…is also important. Some studies, some researchers or else? Important for what? Please specify.

Our reply 6:

We have revised the sentence as follows:

Some studies suggest that subjective SES is also important for ensuring sound physical and mental health among older adults.

Editor comment 7: 

Background:

- The last two sentences can also be improved. The authors stated that “no studies have been conducted on the impact of both objective and subjective SES on depressive symptoms in older adults in Myanmar.” Then the objective read, “…whether subjective or objective SES is associated with depressive symptoms in older adults in Myanmar.” The expression redundancy makes the reading quite uncomfortable.

Our reply 7:

We have revised the sentence as follows:

However, few studies have been conducted on the impact of both objective and subjective SES on mental health among older adults. This study examines whether objective or subjective SES is associated with depressive symptoms in older adults in Myanmar. 

Editor comment 8: 

Methods:

- Please revise the first sentence to read that you conducted the study using ‘a multistage random sampling method’ to recruit the participants and ‘face-to-face interviews’ to collect the data. The current statements mean you conducted the sampling methods and face-to-face interviews, which is not entirely correct. Please also provide essential information in this section (study period, design, and participants). Consider something like, “This cross-sectional study was conducted in… A multistage sampling method was used to recruit participants from two regions for face-to-face interviews.” No need to repeat ‘in Myanmar.”

Our reply 8:

We have revised the sentence as follows based on your suggestion.

This cross-sectional study was conducted between September and December, 2018. A multistage sampling method was used to recruit participants from two regions of Myanmar, for face-to-face interviews. 

Editor comment 9:

Methods:

Please summarize the variable measurements and keep space for briefly describing statistical methods.

Our reply 9:

We have revised and added sentences as follows:

Objective and subjective SES were assessed by using the wealth index and asking participants a multiple-choice question about their current financial situation, respectively. The relationship between objective/subjective SES and depressive symptoms was examined using a multivariable logistic regression analysis.

Editor comment 10:

Methods:

Please move the number of participants and age to results.

Our reply 10:

We have moved the number of participants and age to the results section.

Results

The mean age of the 1,200 participants aged 60 years and above was 69.7 (SD: 7.4), and 706 (58.8%) were female. Among them, 265 (22.1%) had depressive symptoms. 

Editor comment 11: 

The results section is too slim as compared to other sections:

- The authors may provide the number of participants included in the analyses with a mean age (or median) and SD (or IQR) and sex.

- Please also provide the proportion of participants with depressive symptoms.

- Were the results in the first sentence from bivariate analyses? If so, please state it clearly. Alternatively, the authors may present only multiple regression results in the abstract.

- Please provide statistical values to the result in the last sentence.

Our reply 11:

We have revised the Results section as follows:

The mean age of the 1,200 participants aged 60 years and above was 69.7 (SD: 7.3), and 706 (59.5%) were female. Among them, 265 (22.3%) had depressive symptoms. After adjusting for objective SES and other covariates, only low subjective SES was positively associated with depressive symptoms (adjusted odds ratio, AOR: 4.18, 95% confidence interval, CI: 2.98–5.87). This association was stronger among participants in the rural areas (urban areas, AOR: 2.10, 95% CI: 1.08–4.05; rural areas, AOR: 5.65, 95% CI: 3.69–8.64).

Editor comment 12:

Conclusions:

- The term ‘influence’ is not appropriate to describe findings of cross-sectional studies that can only tell associations.

- Also, please specify your study participants in interpreting your results.

Our reply 12:

We have revised the sentence as follows:

Subjective SES has a stronger association with depressive symptoms than objective SES among older adults of the two regions in Myanmar, especially in the rural areas.

Editor comment 13:

Introduction

The second sentence (lines 57-59): Previous studies have shown that… Also, when mentioning ‘health,’ please specify if it was ‘physical’ or ‘mental health.’

Our reply 13:

We have revised it as follows:

Previous studies have recognized that low objective SES is correlated with poor physical and mental health in older adults (4-8).

Editor comment 14:

Introduction

Line 59: It is also related to daily care needs (7) and long-term care needs (4, 5, 9).

Our reply 14:

We have revised it as follows:

It also relates to daily care needs and long-term care needs (4, 5, 7, 9).

Editor comment 15:

Introduction

Lines 65-68: The sentence is too long, particularly the part in parenthesis. The references (1-3) can be moved to join those at the end of the sentence. The authors may break the sentence into two.

Our reply 15:

We have revised the sentences as follows:

Subjective SES is defined as a person’s conception of his or her position compared with that of others. For example, if a respondent rated him/herself as lower than middle class in the country or the community, the respondent is defined as having a lower SES (1-3, 11-13).

Editor comment 16:

Introduction

The second (lines 65-89) and third (lines 90-116) are too long and difficult to follow. Please consider restructuring them.

Our reply 16:

We have received professional proof-reading by Editage (www.editage.com) and reconstructed the sentences.

Editor comment 17:

Introduction

Lines 111-113: Please justify how the country’s history of military control and internal conflicts are related to the lack of studies and publications on mental health among older adults.

Our reply 17:

We have revised the sentences as follows:

However, effective medical care systems including mental health services are still underdeveloped. Further, few studies regarding the mental health of older adults have been published in Myanmar. These are due to the international isolation of the country under military control for several years, during which the national health investment was also very low (30-32).

Editor comment 18:

Methods

Please change to ‘Materials and methods.’

Our reply 18:

We have changed ‘Methods’ to ‘Materials and methods’ as per your suggestion.

Editor comment 19:

Methods 

Line 124: The target populations were those…

The information on lines 115-116 and 124-125 is somewhat overlapping.

Our reply 19:

We have revised the Introduction section and the “Material and methods” sections, based on your comments, as follows:

This study aimed to investigate whether objective/subjective SES, which are examined in the same model, are associated with depressive symptoms in older adults in two regions of Myanmar.

The target populations were those in the urban area of the Yangon region and the rural area of the Bago region, 91 kilometers northeast of Yangon.

Editor comment 20:

Methods

The discussion on the representativeness of study samples (lines 126-128) is not in the right place and should be moved to the limitations section.

Our reply 20:

We have moved it to the limitation section as follows:

Third, it is unknown whether these findings are generalizable beyond the Yangon and Bago regions of Myanmar. Myanmar is composed of seven regions and seven states. Therefore, it is difficult to generalize the study findings in Myanmar. However, we may be able to estimate situations of older adults in other regions by the level of urbanization of the selected regions. Ideally, this survey should be extended to include all surrounding regions in the future.

Editor comment 21:

Methods

Line 129: A multistage random sampling method was used to select participants from the two regions.

Line 131:… using a probability proportional to size sampling method?

Our reply 21:

We have revised the sentences following your comments and added a reference.

A multistage random sampling method was used to recruit participants from the two regions. There are 45 townships in the Yangon region and 28 in the Bago region. First, six townships were randomly selected from each region, based on population proportionate sampling (33).

Editor comment 22:

Methods

Line 134: What did ‘again’ mean?

Line 135: What did ‘extracted’ mean?

Our reply 22:

We deleted ‘again’ and ‘extracted’ as follows.

Following this, in Yangon, 10 wards were further randomly selected from each township, while in Bago, 10 village tracts were selected from each township, based on the population of each township/village tract. Finally, 10 people were randomly selected from each ward/village tract using the ledger lists of residents aged 60 years or older.

Editor comment 23:

Methods

Line 136:…aged 60 years or older.

Line 150: In total, 600 people…

Our reply 23:

We have revised them following your comments

Finally, 10 people were randomly selected from each ward/village tract using the ledger lists of residents aged 60 years or older.

In total, 600 people each from the Yangon (222 men and 378 women) and Bago regions (261 men and 339 women) were surveyed.

Editor comment 24:

Methods

Line 154, 182, 189, 206: Please correct the subtitles, which are not understandable.

Our reply 24:

We have deleted ‘Study tool (1), (2), (3) & (4)’, and replaced them as follows based on your comment: 

‘Questionnaire’, ‘Dependent variable’, ‘Independent variables’, and ‘Sociodemographic characteristics’

Editor comment 25:

Methods

Line 155: …based on a questionnaire used in the…

Our reply 25:

We have revised it based on your suggestion.

A 14-page structured questionnaire—based on a questionnaire used in the “Japan Gerontological Evaluation Study” (JAGES)(34)—was developed for face-to-face interviews.

Editor comment 26:

Methods

Lines 160-162: Consider revise it as follows: “The questionnaire was developed in English, translated into Burmese, and back-translated to English to ensure clarity and consistency.”

Our reply 26:

We have revised the sentence following your comment.

Editor comment 27:

Methods

Lines 163-164: “…a group with experiences in conducting epidemiological surveys in Myanmar.” The following sentence (The interviewers were recruited from the company) should be removed.

Our reply 27:

We have revised and removed the sentences as follows, based on your comment:

We hired research staff from the Myanma Perfect Research Company, a group with considerable experience in conducting epidemiological surveys in Myanmar. Before the commencement of the actual survey, a two-day training course on the research protocol, administration of the questionnaire, and ethical concerns was conducted for the interviewers.

Editor comment 28:

Methods

Line 168: Remove ‘small’ from ‘pilot study.’

Our reply 28:

We have removed ‘small’ as follows:

A pilot study was carried out before the actual survey for face validity in Urban Health Center, of the Dagon township, in Yangon.

Editor comment 29:

Lines 169-170: Participants were older adults aged 60 years or older who came to the center's out-patient clinic.

Lines 170-17: We recruited 25 respondents who gave consent to participate in the pilot study in June 2018.

Our reply 29:

We have revised the sentences following your suggestions.

Editor comment 30:

Methods

Line 173: …of the questionnaire?

Our reply 30:

We have revised following your instruction.

During the pilot study, the interviewers ensured sequence, flow, and clarity of the questionnaire.

Editor comment 31:

Methods

Lines 177-180: Please move the inclusion criteria to the earlier section under the ‘Participants.’ Also, the second sentence should read, “We excluded individuals who were bed-ridden or had severe dementia.

Our reply 31:

We have moved it to the earlier section and revised the second sentence following your comment.

The inclusion criteria were individuals aged 60 years or older who were residents of the selected ward or village tract. We excluded individuals who were bed-ridden or had severe dementia. Severe dementia was defined with an Abbreviated Mental Test score of ≤ 6 (34, 35).

Editor comment 32:

Methods

Line 184: Please clarify if the scale was previously validated in Myanmar or other countries?

Our reply 32:

We have clarified it as follows.

We assessed depressive symptoms using the 15-item version of the Geriatric Depression Scale (GDS), which was validated previously in other countries, including Asian countries (38-41).

Editor comment 33:

Methods

Line 191: a principal component analysis was performed…

Line 196: The principal component score was calculated…

Lines 199-200: The participants were asked to select from five options. Their perception of their current financial situation was…

Our reply 33:

We have revised the sentences following your comments.

A principal component analysis was performed on the asset items (e.g., radio, black & white television, color television, Video/DVD player, electric fan, refrigerator, computer, store-bought furniture, personal music player, washing machine, gas cooker, electric cooker or rice cooker, air conditioner, bicycle, motorcycle, van/truck, microwave oven, mobile telephone, and internet).

The principal component score was calculated based on the participants’ possession of each item and used as the wealth index. Subjective SES was assessed by asking: “Which of the following best describes your current financial situation in light of general economic conditions?” The participants were asked to select from five options. Their perception of their current financial situation was 1. very difficult, 2. difficult, 3. average, 4. comfortable, and 5. very comfortable.

Editor comment 34:

Methods

Lines 206: Please consider better terminology to replace ‘Confounding variables.’ I don’t they are not all absolute confounders. ‘Sociodemographic characteristics?’

Our reply 34:

We have revised it to ‘Sociodemographic characteristics’

Editor comment 35:

Methods

Line 209: What does ‘monastic’ mean?

Our reply 35:

We have revised it as follows:

…educational level (no school, the Buddhist monastic school, some/all primary school, middle/high school, or higher)…

Editor comment 36:

Methods

Line 214: …The questions included (1) Do you listen to…; (2)…

Our reply 36:

We have revised the sentences following your comment.

The questions included: (1) Do you listen to someone else’s concerns and complaints? (giving socioemotional support); (2) Do you take care of someone who is sick? (giving instrumental social support); (3) Do you have someone who listens to your concerns and complaints? (receiving emotional social support); and, (4) Do you have someone who takes care of you when you are sick? (receiving instrumental social support).

Editor comment 37:

Methods

Line 225: Change the subtitle to ‘Statistical analyses.’

Line 226: Sociodemographic characteristics were compared…

Line 231: …in bivariate analyses…

Line 232: …were simultaneously entered in a model. Also, please remove ‘using multivariable logistic regression.’

Our reply 37:

We have changed the sentences following your comments. 

Editor comment 38:

Methods

Please use consistent terminologies: ‘multiple’ ‘multivariable,’ or ‘multivariate’

Our reply 38:

We have used ‘multivariable’ as the consistent term.

Editor comment 39:

Methods

Lines 232-233: Adjusted odds ratios (AOR) were presented with 95% confidence intervals (CI).

Our reply 39:

We have changed the sentence following your comment. 

Editor comment 40:

Methods

Further descriptions of data analyses are required to reflect the analyses actually conducted (e.g., stratified by region, sex, etc.). These were not understood until the reader goes through the results.

Our reply 40:

We have added the following sentence.

After performing the analysis on all the subjects, we also performed a stratified analysis by gender and region.

Editor comment 41:

Results

Lines 259-262: The results of the living status are interesting. The authors reported that more than 90% of the participants lived with others. In several cultures, living with others refers to living people other than their family or relatives (negative). However, in the table, the authors indicated two categories of the living status – ‘alone’ and ‘not alone.’ I assumed that living with others in this context means living with family or relatives (positive, with more social support). This is confirmed by the comparative results showing that the proportion of participants who did not live alone was significantly higher in those who did not have depressive symptoms. I would suggest changing the term ‘living with others’ to ‘not living alone.’

Our reply 41: 

We have changed the term ‘living with others’ to ‘did not live alone’ following your comment.

Most of the participants did not live alone (94.4%), and the rate of participants who did not live alone was significantly higher among those who did not have depressive symptoms (96.0%) than those who had depressive symptoms (88.7%).

Editor comment 42:

Results

Lines 273-278: The sentence is too long and hard to read. Consider revising it: “Depressive symptoms were positively associated with being female (AOR: 1.64, 95% CI: 1.15–2.34), experiencing illness during the preceding year (AOR: 1.92, 95% CI: 1.41–2.61), and living in Bago (AOR:1.62, 95% CI: 1.10–2.38). Depressive symptoms were negatively associated with receiving instrumental support (AOR: 0.31, 95% CI: 0.12–0.77) and frequency of visits to religious facilities once per week or more (AOR: 0.57, 95% CI: 0.42–0.77).”

Our reply 42:

We have revised the sentences following your comment.

Editor comment 43:

Results

Line 278: “Even after adjusting for these confounding variables,…” brings more confusion than help. In my understanding, the results in this section are from multiple regression (adjusted). Suggest removing this or stating this at the beginning of the paragraph. 

Lines 280-281: “…., but low objective SES was not significantly associated (AOR: 0.90, 95% CI: 0.61–1.32)” is incomplete. Suggest removing it as it is not statistically significant.

Our reply 43:

We have removed the following: “Even after adjusting for these confounding variables” & “…, but low objective SES was not significantly associated (AOR: 0.90, 95% CI: 0.61–1.32),”

Low subjective SES was positively associated with depressive symptoms (AOR: 4.18, 95% CI: 2.98–5.87) (Table 2).

Editor comment 44:

Results

Lines 281-284: Like the above comment, please make the comparisons expressions complete; e.g., still significant, low objective SES was not. 

Our reply 44:

We have completed the sentence which was significantly associated with depressive symptoms. We also have removed the sentence which was not significantly associated with depressive symptoms following your comment.

Low subjective SES was still significantly associated with depressive symptoms after being stratified by region—Yangon (AOR: 2.10, 95% CI: 1.08–4.05) and Bago (AOR: 5.65, 95% CI: 3.69–8.64) (Table 2a).

Editor comment 45:

Results

Lines 285-289: Suggest removing all non-significant results from the text. If needed, the authors may summarize them in a sentence and refer the reader to the tables.

Line 289: The expression ‘Meanwhile’ sounds awkward in the sentence as the earlier results were not statistically significant, and the following were.

Lines 289-301: Please rewrite the sentences, breaking them into smaller and simpler sentences as they are complicated to read.

Our reply 45:

We have revised the sentences following your comments.

Among the men in Yangon, receiving instrumental support and the frequency of visits to religious facilities once per week or more were both negatively associated with depressive symptoms (receiving instrumental support: AOR: 0.05, 95% CI: 0.01–0.38; frequency of visits to religious facilities: AOR: 0.23, 95% CI: 0.05–0.96) (Table 2a). 

Among the residents of Bago, low subjective SES was positively associated with depressive symptoms among both men and women (men: AOR: 8.97, 95% CI: 4.46–18.07; women: AOR: 4.45, 95% CI: 2.57–7.72). On the other hand, the frequency of visits to religious facilities once per week or more was negatively associated with depressive symptoms (rural men: AOR: 0.45, 95% CI: 0.23–0.87, rural women: AOR: 0.39, 95% CI: 0.23–0.67) (Table 2b). Variables that were not significantly associated with depressive symptoms can be found in Tables 2, 2a and 2b.

Editor comment 46:

Discussion

Lines 310-311: The discussion needs further elaboration – what were the results differences?

Our reply 46:

We have revised the sentence as follows:

Previous studies in middle- or high-income countries only observed a significant association between objective SES and mental health [5, 6, 9]. In our study, however, those with low subjective SES were more likely to experience depressive symptoms than those with average or higher subjective SES, even after adjusting for potential confounding factors. 

Editor comment 47:

Discussion

Lines 312-313: How this study’s result was similar to the results of a previous meta-analysis showing that subjective SES affects physical health more than objective SES? Did this study also examine the association between subjective SES and physical health? Also, please clarify the settings of studies included in the meta-analysis – Lower- and middle-income or high-income countries?

Line 314: “… this is the first study showing similar results for mental health.” This sentence is not understandable – similar to what? Mental health and what?

Our reply 47:

Based on your comments, we have revised the sentences as follows:

Although a previous meta-analysis in middle- or high-income countries did not investigate the association between SES and mental health, it showed that subjective SES affects physical health more than objective SES [19]. To the best of our knowledge, this is the first study showing that subjective SES is also associated with mental health more than objective SES. 

Editor comment 48:

Discussion

Lines 314-316: Consider simplifying this sentence: “Moreover, we found that low subjective SES was associated with depressive symptoms in rural older adults, but not in urban older adults.”

Our reply 48:

We have revised the sentence following your comment.

Editor comment 49:

Discussion

Lines 318-319: Please clarify ‘…appeared to have depression’ and ‘have indications of depressive symptoms.’ Are these equivalent to ‘having depressive symptoms?’ If not, the prevalence rates shouldn’t be compared. The authors may have to go to the original scales or measures used in those studies.

Our reply 49:

Yes, these are equivalent to ‘having depressive symptoms’. 

We have revised the sentence as follows:

Previous research on community-dwelling older adults in Myanmar indicated that 22.2% of them have depressive symptoms [26]. A large-scale survey on older adults conducted in Myanmar also revealed that approximatly 16% to 56% have depressive symptoms [27]; 22.1% of the present study falls within this category.

Editor comment 50:

Discussion

Line 325: What is the global prevalence? Please cite the sources.

Our reply 50:

We have added the reference as follows:

Although we cannot directly compare the prevalence in this study with that of previous studies because of differences in the study period, the measure of depressive symptoms used, and the inclusion and exclusion criteria for the study population, the prevalence in Myanmar was relatively higher than the global prevalence [50].

Editor comment 51:

Discussion

Line 328: As commented earlier, please be specific when using the terminologies. It may not be accurate to say ‘confounding factors’ when the authors based only on the available variables associated with the outcome variable in bivariate analyses. A confounding variable must be associated with both independent and dependent variables. Suggest simply say something like “other covariates in the multiple regression model.”

Our reply 51:

We have revised the sentence following your comment.

Although both low objective and subjective SES were significantly associated with depressive symptoms by a bivariate analysis, only subjective SES was associated after adjusting for objective SES and other covariates in the multiple regression model.

Editor comment 52:

Discussion

Lines 326-348: Please consider re-structuring the paragraph, which is too long and difficult to read. Do not discuss different results in one paragraph. For example, I cannot see the connection between the first and second paragraphs.

Our reply 52:

We have revised the sentence and added sentences to show the connection.

Although both low objective and subjective SES were significantly associated with depressive symptoms by a bivariate analysis, only subjective SES was associated after adjusting for objective SES and other covariates in the multiple regression model. Similar to other covariates, sex and physical illness were associated with depressive symptoms in this study. It is well known that women are generally more likely to be depressed than men [51] and that physical illness is associated with depressive symptoms [52]. 

Editor comment 53:

Discussion

Lines 331-332: I cannot get what this sentence says, “However, the impact of low subjective SES was more than two times stronger than these variables.” How was ‘impact’ measured in a cross-sectional study? What are ‘these variables?’

Our reply 53:

We have revised the sentences as follows.

However, the AOR of low subjective SES compared with middle/high subjective SES was greater than that of the AORs of sex and physical illness.

Editor comment 54:

Discussion

Lines 332-348: The authors spent too much space discussing the country’s health disparities and inequitable resource allocation. This information is important; however, it does not help explain the direct relationships between SES and mental health. The literature on the pathways from health disparities and mental health may help.

Our reply 54:

We have reduced the discussion about the country’s health disparities and inequitable resource allocation. We have left the references to literatures, nonetheless. 

Persistent inequalities exist in health outcomes in Myanmar's seven states and seven regions [53, 54]. According to Zaw et al. [54], conventional budget allocations related to population and infrastructure provide disproportionately more resources to regions with better health and less resources to areas with high health needs in Myanmar. Even in Japan, considered an egalitarian society—as reflected by a Gini coefficient of 63% in 2019 [55]—with relatively few inequalities in health [56], substantial social inequalities in mental health, measured by SSS, were identified [11].

Editor comment 55:

Discussion

Lines 352-356: Please consider refining the long sentence. Also, the authors should focus on the relationships between SES and mental health, not physical health. The mechanisms and pathways to the outcomes can be different.

Our reply 55:

We have refined the long sentence and focused on the relationships between SES and mental health. 

Another reason may be related to negative psychological consequences through stress-related psychological pathways due to low subjective SES. The idea is supported by empirical evidence showing that low SSS is associated with higher physiological stress markers [57-59]. Evidence also suggests that the stress-related dysregulation of the hypothalamic–pituitary–adrenal (HPA) axis, a part of the neuroendocrine system controlling responses to stress, predicts the onset and recurrence of depression [60, 61]. From these perspectives, the neuroendocrine pathway may link low subjective SES to depressive symptoms.

Editor comment 56:

Discussion

Lines 378-380: I am not sure if this statement supports the finding of the relationship between religiosity and mental health. While religiosity was higher in older adults and people living in rural areas, this study also shows that depression was also high among them.

Our reply 56:

Yes, you are exactly correct in your perception.

Thus, we have discussed it as follows:

There is a possibility that religiousness mediates the relationship between subjective SES and depressive symptoms for rural older adults in Myanmar, who may have been at a much greater risk of developing depressive symptoms if they had not been religious.

Editor comment 57:

Discussion

Lines 384-434: The study’s limitations section is excessive and presented in a giant paragraph, which is very difficult to follow. Please condense and restructure it, reducing at least 70% of the contents.

Our reply 57:

We have restructures and reduced the paragraph, in length. However, we have not removed the parts which reviewers previously pointed out. 

Editor comment 58:

Discussion

Line 384: Please clarify the data collection method and make it consistent across the paper – face-to-face or self-administered?

Lines 384-390: In the first limitation, the authors discussed two different issues – self-report measures and depressive symptoms measurement. These issues should not come along together.

Our reply 58:

We have revised it following your comments: 

First, the nature of the face-to-face interviews did not allow for the objective assessment of participants’ situations [69]. The assessment may have caused social desirability bias, resulting in misreporting of depressive symptoms. Second, our measurement of depressive symptoms was based only on the GDS scores, without corroborating clinical evaluation. However, the GDS is a validated instrument for assessing depressive symptoms and is used widely [38, 40, 41, 48, 70].

Editor comment 59:

Discussion

Lines 391-397: I am not sure if the explanation on the disparities in health care access is really helpful, while this study looked at the relationship between SES and depressive symptoms.

Our reply 59:

We have deleted the explanation on the disparities in health care access, and have revised the sentence as follows:

Third, it is unknown whether these findings are generalizable beyond Yangon and Bago. Myanmar is composed of seven regions and seven states. Therefore, it is difficult to generalize the study findings to the population in Myanmar. However, we may able to estimate situations of older adults in other regions by the level of urbanization of the selected regions.

Editor comment 60:

Discussion

Lines 403-422: The authors spent 19 lines to discuss one limitation. This can be reduced at least by 80%. This is also true for the final limitation (lines 422-434)

Our reply 60:

We have reduced the sixth limitation and final limitation as follows:

Sixth, indicators for bonding social capital, which are derived from relationships between similar persons such as with respect to sociodemographic and socioeconomic characteristics, were not included in our analysis. Previous research suggested that rural areas are richer than urban areas in bonding social capital [71, 72]. Although we adjusted for instrumental social support, which was associated with depressive symptoms in a bivariate analysis, further studies are needed to examine the association between types of social support and human interactions and depressive symptoms, in Myanmar. Finally, we developed the questionnaire in the dominant language based on the “Linguistic Validation Manual for Health Outcome Assessment [37].” However, it is necessary to improve the accuracy of the questionnaire to minimize such errors and obtain higher-quality results.

Editor comment 61:

Discussion

Lines 435-440: Not sure what the paragraph is about – limitations or recommendations? If it is the latter, it should be integrated with the conclusions.

Line 441-444: Overall, the conclusion is weak and hard to understand. The term ‘influence’ is not appropriate for describing cross-sectional associations. It’s also difficult to tell what the last sentence is about – was it based on this study or from other studies or the general perception (who considered it in that way?).

The authors may conclude the paper by clearly summarizing the key findings, followed by brief recommendations for policy, programs, or future studies.

Our reply 61:

We have integrated and revised the conclusion based on your comments. 

In conclusion, the association between subjective SES and depressive symptoms have been greater than that of objective SES and depressive symptoms in the two urban and rural areas in Myanmar, especially in the rural area. Considering not only material wealth, but also to subjective SES should be important for decreasing depression in older adults in the area. Intervention programs for depression in older adults, which include social protection, sustainable livelihood, and wealth creation, should also consider regional differences in the context of subjective SES by reducing economic disparities between rural and urban areas and within communities. A detailed study should also be conducted on how unique factors of the cultural background such as religiosity affect the mental health of older men and women in Myanmar.

Editor comment 64:

References

Please revise the references list to be consistent and aligned with the journal’s guidelines.

Our reply 64:

We have revised the references list using PLoS style, based on the journal’s guidelines to use the Vancouver style. 

https://journals.plos.org/plosone/s/submission-guidelines#loc-references

---

## [Editor Report · Decision Letter 3]

4 Jan 2021

Association between depressive symptoms and objective/subjective socioeconomic status among older adults of two regions in Myanmar

PONE-D-20-15915R3

Dear Dr. Sasaki,

We’re pleased to inform you that your manuscript has been judged scientifically suitable for publication and will be formally accepted for publication once it meets all outstanding technical requirements.

Kind regards,

Siyan Yi, MD, MHSc, PhD

Academic Editor

PLOS ONE

---

## [Editor Report · Acceptance letter]

14 Jan 2021

PONE-D-20-15915R3 

Association between depressive symptoms and objective/subjective socioeconomic status among older adults of two regions in Myanmar 

Dear Dr. Sasaki:

I'm pleased to inform you that your manuscript has been deemed suitable for publication in PLOS ONE. Congratulations! Your manuscript is now with our production department. 

Kind regards, 

on behalf of

Dr. Siyan Yi 

Academic Editor

PLOS ONE